# Future Perspectives on the TNM Staging for Lung Cancer

**DOI:** 10.3390/cancers13081940

**Published:** 2021-04-17

**Authors:** Ramón Rami-Porta

**Affiliations:** 1Department of Thoracic Surgery, Hospital Universitari Mútua Terrassa, University of Barcelona, Plaza Dr. Robert 5, 08221 Terrassa, Spain; rramip@yahoo.es; 2Network of Centers for Biomedical Research in Respiratory Diseases (CIBERES) Lung Cancer Group, 08221 Terrassa, Spain

**Keywords:** biomarkers, lung cancer, lung cancer staging, TNM classification

## Abstract

**Simple Summary:**

The tumor, lymph node, and metastases classification of lung cancer describes how big the tumor is and what structures it invades; whether the lymph nodes are involved or not, and, if they are involved, where they are in the chest; and whether there are tumor implants in other organs distant from the lung. This classification of lung cancer was proposed in 1966 and it has undergone periodic revisions. The revisions for its 7th and 8th editions were based on a large amount of international data from patients treated with different therapeutic modalities. New data are being registered now to inform the forthcoming 9th edition. A potential innovation of this edition will be the combination of this classification with other prognostic factors, such as mutations of the genetic code of the cancer cells among others, to achieve a more personalized prognosis for individual patients.

**Abstract:**

Since its conception by Pierre Denoix in the mid-20th century, the tumor, node, and metastasis (TNM) classification has undergone seven revisions. The North American database managed by Clifton Mountain was used to inform the 2nd to the 6th editions, and an international database collected by the International Association for the Study of Lung Cancer, promoted by Peter Goldstraw, was used to inform the 7th and the 8th editions. In these two latest editions, it was evident that the impact of tumor size was much greater than it was suggested in previous editions; that the amount of nodal disease had prognostic relevance; and that the number and location of the distant metastases had prognostic implications. However, the TNM classification is not the only prognostic factor. Data are being collected now to inform the 9th edition of the TNM classification, scheduled for publication in 2024. Patient-, environment-, and tumor-related factors, including biomarkers (genetic biomarkers, copy number alterations, and protein alterations) are being collected to combine them in prognostic groups to enhance the prognosis provided by the mere anatomic extent of the tumor, and to offer a more personalized prognosis to an individual patient. International collaboration is essential to build a large and detailed database to achieve these objectives.

## 1. Introduction

Pierre Denoix (1912–1990), a surgical oncologist from France, developed the classification of anatomic tumor extent in a series of articles that he published between 1943 and 1952. He based this classification on three components: the extent of the primary tumor (T), the nodal spread (N), and the distant metastases (M)—the TNM classification. The Union for International Cancer Control (UICC) adopted this system and published brochures of the TNM classifications of individual tumors—the one for lung cancer appeared in 1966. After two years, the UICC published the first manual of the TNM classification of malignant tumors. The American Joint Committee on Cancer (AJCC) also adopted this classification and published its first manual in 1977. Since then, the UICC and the AJCC are in charge of promulgating and revising the TNM classification and staging system of malignant tumors. Since the 5th edition of the classification, published in 1997, both agencies have revised the classification and published their respective manuals in unison [1]. Table 1 shows the timeline of the different editions of the TNM classification for lung cancer.

A North American database, managed and analyzed by Clifton Mountain (1924–2007), a thoracic surgeon from the United States of America, was used to inform the 2nd to the 6th editions. This database consisted of 5319 patients from North America. Most of them had undergone surgical treatment of the disease, although the tumors had both clinical and pathologic classification. In order to overcome the intrinsic limitations of this database regarding number of cases, geographic representation, and variety of therapies, Peter Goldstraw, a thoracic surgeon from Great Britain, proposed in 1996 the creation of a Staging Committee within the International Association for the Study of Lung Cancer (IASLC) with the main objective to build a solid international database, including patients from as many countries as possible and treated with all therapeutic modalities, that could be used to revise future editions of the TNM classification for lung cancer [2].

The IASLC Staging Committee, renamed IASLC Staging and Prognostic Factors Committee (SPFC) in 2013, was established in 1998 and already has completed two phases of the IASLC Staging Project. The objective of the first phase of the project (1998–2009) was to update the 6th edition of the TNM classification of lung cancer. A database of 81,495 evaluable patients with lung cancer, diagnosed from 1990 to 2000 and submitted from 45 sources in 20 countries, was used for that revision [3]. From the analyses of these data, several recommendations for changes in the 6th edition TNM emerged, were accepted by the UICC and the AJCC, and eventually led to the 7th edition of the classification [4].

In the second phase of the IASLC Staging Project (2009–2016), malignant pleural mesothelioma and epithelial thymic tumors were added to the project, and an agreement was made with the Worldwide Esophageal Cancer Collaboration (WECC) to disseminate the forthcoming 8th edition TNM of esophageal and esophagogastric juncture cancers together with the respective TNM classifications of the other thoracic malignancies. For the revision of the 7th edition TNM of lung cancer, 77,156 patients diagnosed from 1999 to 2010, submitted from 35 sources in 16 countries, were registered and evaluable [5]. The recommendations for changes generated from the multiple analyses performed were also accepted by the UICC and the AJCC, and resulted in the 8th edition of the TNM classification of lung cancer [6].

The objective of this Perspective article is to highlight the innovations introduced in the 7th and 8th editions TNM of lung cancer and to discuss what can be expected from the 9th edition of the classification due to be published in 2024.

## 2. Data-Based Innovations of the 7th and the 8th Edition of the TNM Classification

The innovations introduced in the 7th and the 8th editions of the TNM classification of lung cancer represent the progressive acquisition of knowledge based on the analyses of large, international databases.

### 2.1. Tumor Size: Every Centimeter Counts

It was already evident in the analyses of tumor size for the 7th edition that size mattered more than it had been reflected in the previous editions of the TNM classification. While the 3 cm landmark still separated T1 from T2 tumors, there was enough evidence to divide T1 into T1a (≤2 cm) and T1b (>2–3 cm); and T2 into T2a (>3–5 cm) and T2b (>5–7 cm); and to reclassify tumors > 7 cm as T3 based on the differences in prognosis [7]. The importance of tumor size was even more evident in the analyses that led to the 8th edition. Tumors of ≤2 cm could be subdivided into those of ≤1 cm (the new T1a) and those of >1–2 cm (the new T1b). A new subcategory was added for those > 2–3 cm (the new T1c). It was also evident that the T2 category could also be refined, and the 7th edition T2a and T2b were redefined as tumors > 3–4 cm and >4–5 cm, respectively; those > 5–7 cm were redefined as T3; and those > 7 cm, as T4, in the 8th edition [8].

The TNM classification requires the registration of the largest dimension to assign a T category based on tumor size, but it never gave any particular instruction on how to measure it. For the first time in the history of the classification, the 8th edition provided clinicians and pathologists with recommendations on how to measure tumor size in three different clinical situations. (1) For solid tumors, it is recommended to use the lung window of the computed tomography in the projection that reveals the largest tumor dimension. (2) For part-solid non-mucinous adenocarcinomas with ground glass features on the computed tomography and lepidic growth on pathologic examination, the size to be registered is that of the solid component on computed tomography for the clinical T category, and the size of the invasive component on pathologic examination for the pathologic T category. (3) The measurement of the pathologic tumor size can be difficult after induction therapy, if there is objective tumor response. If only are there viable cells, the recommendation is to multiply the percent of viable cells by the size of the residual tumor mass [9].

### 2.2. Prognostic Relevance of Visceral Pleura Invasion

In the 7th edition, visceral pleura invasion was defined as tumor involvement of its elastic layer. The involvement of the innermost part of the visceral pleura without reaching its elastic layer is coded PL0 and does not count as a T descriptor. The involvement of the elastic layer is coded PL1 and classifies a tumor of ≤3 cm as T2a. When tumor invasion involves the lung surface, it is coded PL2 and also is a T2 descriptor [10]. The analyses of the database used for the 8th edition confirmed that visceral pleural invasion was properly assigned to the T2 category, but also showed that PL2 had significantly worse prognosis that PL1 [8]. Given the prognostic relevance of the invasion of the visceral pleura, the use of elastic stains was recommended in the 7th edition when the elastic layer is not clearly seen on hematoxylin-eosin, and this recommendation was further emphasized in the 8th edition.

The invasion of the visceral pleura is associated with worse prognosis. Shimizu et al. found that, among 1074 patients with pathologic (p) T1 and pT2 non-small cell lung cancers, 288 (26.8%) presented with visceral pleura invasion. The 5- and 10-year survival rates for those with and without visceral pleura invasion were 49.8 and 37.0%, and 76.0 and 53.2%, respectively (*p* < 0.0001). The worse prognosis associated with the invasion of the visceral pleura occurred regardless of nodal status, but those tumors with visceral pleura invasion had a higher rate of nodal disease [11]. The same group of authors analyzed the impact of the type of visceral pleura invasion and tumor size on prognosis, and found that, for tumors 3 cm or less, there were no differences between PL0 and PL1, but PL2 had worse prognosis than PL0. For tumors greater than 3 cm, the prognosis of PL1 or PL2 was similar to that of T3 tumors [12]. Therefore, they proposed to upstage tumors larger than 3 cm with visceral pleura invasion to the T3 category. Yoshida et al., in a very detailed analysis of the prognostic impact of visceral pleura invasion and tumor size based on 9758 patients collected by the Japanese Joint Committee for Lung Cancer Registration in 1999, concluded that the T category of tumors 7 cm or less in greatest dimension should be upstaged to the next T category, if they had visceral pleura invasion [13]. In the study of Sakakura et al., 5-year survival rates after resection of 654 patients with 8th edition T2a-PL1 tumors, 274 with T2a-PL2 tumors, and 219 with T2b tumors were 76.3, 54.6, and 66.6%, respectively. While there were statistically significant differences between T2a-PL1 and T2a-PL2 (*p* = 0.001), survival was similar for T2a-PL2 and T2b (*p* = 0.160) [14]. So, they proposed to classify T2a-PL2 tumors as T2b.

Another important issue is the invasion of the adjacent lobe by the primary tumor. Okada et al., based on 19 cases of invasion of the adjacent lobe among 132 patients who had undergone resection for T3 non-small cell lung cancers, found that the invasion of the adjacent lobe had worse prognosis than that of T1 and T2 tumors, but prognosis was similar to that of T3. Based on these results, they concluded that the involvement of the adjacent lobe should be classified as T3 [15]. Ohtaki et al. analyzed the survival of 90 (4.3%) patients with adjacent lobe invasion among 2097 patients operated on for lung cancer. They differentiated two types of invasion: direct, that is, the involvement of the adjacent lobe when the lung fissure is incomplete; and across, meaning the involvement of the adjacent lobe when there is a complete fissure. The 5-year survival rate of the 90 patients with adjacent lobe invasion was 59.6%, between that of T2a (61.0%) and T2b (49.2%). However, the 5-year survival rate for those patients with direct invasion was significantly better than that of patients with adjacent lobe invasion across a complete fissure: 85.7 and 52.0%, respectively (*p* = 0.010). They concluded that those tumors 5 cm or less in size with invasion of the adjacent lobe across an incomplete fissure should be upstaged to the 7th edition T2b category, while the direct invasion of the adjacent lobe through an incomplete fissure should not modify the T category [16]. Andreetti et al. analyzed 135 patients with cancers involving the adjacent lobe and also found statistically significant differences between the 5-year survival rates of those patients with direct invasion and with invasion across a complete fissure: 63.9 and 48.9%, respectively (*p* = 0.01). In addition, they found significantly higher rates of nodal disease (35 and 4%, *p* = 0.004) and recurrence (43 and 16%, *p* = 0.01) among patients with invasion across a complete fissure [17].

The best resection when the adjacent lobe is involved still is a matter of debate. Data on the topic are scarce, but Okada et al. did not find significant differences in the survival of 10 patients who had undergone peumonectomy or bilobectomy and 9 who had undergone lobectomy and sublobar resection of the involved adjacent lobe (*p* = 0.9115) [15]. Andreetti et al. did not find statistically significant differences between the 5-year survival rates of pneumonectomy, bilobectomy, lobectomy with segmentectomy, or lobectomy with wedge resection: 39.8, 53.9, 54.3, and 59.8%, respectively (*p* = 0.09) [17]. From these two reports, it seems that a sublobar resection of the involved adjacent lobe could be an adequate oncologic operation.

### 2.3. Separate Tumor Nodules, and Malignant Pleural and Pericardial Effusions, and Nodules

The 7th edition modified the categories of the additional tumor nodules found in the ipsilateral and contralateral lung, and assigned a T3 category to those located in the lobe of the primary tumor; a T4 category to those found in another ipsilateral lobe different from the one of the primary tumor; and a M1a category to those in the contralateral lung [7]. These categories remained unchanged in the 8th edition [8].

Malignant pleural effusions had been classified as T4 in previous TNM editions, although they were treated as advanced disease and had a prognosis similar to metastatic dissemination. In the 7th edition, there were enough data to reassign malignant pleural and pericardial effusion and nodules to the M component, in the M1a category, defined as metastases within the chest cavity [7]. The analyses of the database for the 8th edition confirmed their correct allocation as M1a [8].

### 2.4. Changes in Other T Descriptors

No other changes were made in the 7th edition, but the analyses of the data used to inform the 8th edition showed that some descriptors could be reassigned to different T categories. Total atelectasis and endobronchial location < 2 cm from the carina had been classified as T3 in the 7th edition, but the new analyses showed that their prognosis was aligned with that of T2 descriptors. Therefore, they were reclassified as T2 in the 8th edition. However, the invasion of the diaphragm, that also had been classified as T3, was found to have worse prognosis than that of the other T3 descriptors and similar to those within the T4 category. It was, then, reclassified as T4 in the 8th edition. Finally, the invasion of the mediastinal pleura was deleted as a T3 descriptor because it is seldom found as a single descriptor [8].

### 2.5. Quantification of Nodal Disease

Nodal disease is classified according to the anatomic location of the involved lymph nodes. In other cancers, such as those of the digestive tract, the clinical and pathologic N categories are based on the number of involved lymph nodes. Quantification of nodal disease can be done in several ways: by counting the number of involved lymph nodes, nodal stations, or nodal zones; by dividing the number of involved lymph nodes by the total number of lymph nodes removed —the so called lymph node ratio; and by combining the number of removed lymph nodes and the number of nodal stations involved [18,19].

In the analyses of the 7th edition, the number of nodal zones was used to quantify nodal disease. Nodal zones were defined as groupings of neighboring nodal stations in the same anatomic area. Their main objective was to facilitate nodal staging in those patients who would not undergo surgical resection. The following nodal zones were defined based on the Mountain and Dresler nodal map [20]: upper zone, including the highest mediastinal, the upper paratracheal, the prevascular and retrotracheal, and the lower paratracheal nodal stations; the aortopulmonary zone, including the subaortic and the para-aortic nodal stations; the subcarinal zone, including the subcarinal nodal station, only; the lower zone, including the paraesophageal and the pulmonary ligament nodal stations; the hilar zone, including the hilar and the interlobar nodal stations; and the peripheral zone, including the lobar, segmental and subsegmental nodal stations [21]. Table 2 shows the proposed new categories and their survival.

The four proposed categories form three different prognostic groups (N1a, N1b and N2a, and N2b), because the prognosis of the involvement of multiple N1 zones is similar to that of the involvement of a single N2 zone.

A similar quantification analysis was done for the 8th edition, but that time it was based on the number of involved nodal stations depicted in the lymph node map proposed by the IASLC [22,23]. This map had a new station including three nodal groups: the low cervical, supraclavicular, and sternal notch nodes. The new categories and their survival are shown in Table 3.

Because the prognosis of N2a1 and N1b is similar, these five categories form four prognostic groups: N1a, N1b and N2a1, N2a2, and N2b.

The above quantification analyses were performed on data arising from pathologic staging of tumors that had been resected and had an acceptable intraoperative nodal evaluation. When validation in the clinical staging was attempted, the prognostic differentiation of the proposed categories was not found, because clinical staging is not so accurate as pathologic staging. This was the main reason why these new categories could not be introduced into the official N classification and remain as optional for further analyses. However, they have clinical relevance in those patients with lung cancer who undergo resection and whose tumors are found to have nodal disease: these proposed new categories assist us in refining postoperative prognosis and can be the basis for a more intensified adjuvant therapy or follow-up.

### 2.6. The Number and Location of Metastases Matter

An innovation in the 7th edition TNM was the separation of intrathoracic metastases from extrathoracic metastases that were coded as M1a and M1b, respectively [24]. The analyses of the 8th edition database still allowed further differentiation in the M1b category. It was found that a single extrathoracic metastasis had the same prognosis as intrathoracic metastases, but prognosis was better than that of multiple extrathoracic metastases, either in one or in several organs. According to these findings, the 7th edition M1b was redefined in the 8th edition as the involvement of a single extrathoracic metastasis, including a single involved lymph node beyond the nodal stations included in the pulmonary and mediastinal lymph node chart. A new M1c category was created to include the presence of multiple extrathoracic metastases [25]. In addition, the MX category was deleted by the UICC and the AJCC. It was thought that even by taking a good medical history and performing a physical examination, it is possible to elucidate if a lung cancer has metastatic dissemination or not.

### 2.7. Small Cell Lung Cancer and Bronchopulmonary Carcinoids

In the 7th and the 8th editions of the TNM classification of lung cancer, the first exploratory analyses were performed in the population of patients with non-small cell lung cancer and were later tested in patients with small cell lung cancer. All the analyses on the small cell lung cancer showed that the TNM classification works well, although the survival rates of the different stages are lower than those for their non-small cell counterparts, reflecting the different natural history and biology of both carcinomas. The TNM classification is useful both in the surgical and non-surgical populations, and represents a refinement in prognosis, because the limited disease of the classic dichotomous staging system (limited vs extensive disease) includes TNM stages I, II, and III with statistically significant differences in prognosis [26,27,28].

Although rather limited by the relatively few cases of bronchopulmonary carcinoids and by the fact that they could not be differentiated between typical and atypical, the analyses of the 7th edition showed that the TNM classification also worked for these tumors that had been traditionally excluded from this classification. In this case, the survival curves were much better than those for small and non-small cell lung cancer, reflecting the indolent nature of these tumors even when there is nodal involvement or metastatic disease. Therefore, bronchopulmonary carcinoids were included in the TNM classification of lung cancer for the first time in the history of the classification [29].

### 2.8. Lung Cancer Presenting with Multiple Lesions

Lung cancers presenting with multiple lesions are nowadays a common finding, and pose difficulties regarding their classification. To deal with this problem, these cancers were divided in four patterns of disease. The published evidence was thoroughly reviewed and recommendations were issued for clinical and pathologic diagnosis, and for the homogenous classification of these tumors [30,31,32,33]. Table 4 shows the recommended classification for each pattern of disease.

### 2.9. Newly Defined Adenocarcinomas

In 2011, the IASLC, the American Thoracic Society and the European Respiratory Society defined in a joint effort two new lung cancers: adenocarcinoma in situ and minimally invasive adenocarcinoma. Both are no greater than 3 cm. Adenocarcinoma in situ usually presents as a pure ground glass opacity on high resolution computed tomography, while minimally invasive adenocarcinoma has a predominant ground glass pattern with a solid component no greater than 5 mm. At microscopic examination, adenocarcinoma in situ presents with a pure lepidic growth with no signs of invasion; and minimally invasive adenocarcinoma has a predominant lepidic pattern but with an invasive portion of any histologic subtype not greater than 5 mm [34]. In 2015, these two tumors were included in the official classification of the World Health Organization [35]; and one year later they were given a code in the TNM classification: adenocarcinoma in situ was coded as Tis(AIS); and minimally invasive adenocarcinoma, as T1mi [9]. In Tis(AIS), it is important to add AIS between parenthesis to differentiate it from the already existing squamous cell carcinoma in situ, which is now coded as Tis(SCIS).

## 3. Future Perspectives

At the time of this writing, data collection for the 9th edition of the TNM classification is in progress. Given the rate of submitted cases up to now, there are all reasons to think that the number of cases available for analyses and for informing the 9th edition TNM will be similar to that of the previous two editions. This section is based mainly on the discussions during the meetings of the subcommittees of the IASLC SPFC held prior to the postponed 2020 World Conference on Lung Cancer that took place virtually from 28 to 31 January 2021.

### 3.1. Methodology and Validation

In the 7th and the 8th editions, the T, the N, and the M components of the classification, as well as the stage grouping were dealt with according to the methodology discussed within each of the subcommittees in charge of each component [36,37,38]. For the 9th edition, a Validation and Methodology Subcommittee was created in order to establish, before the beginning of the analyses of the database, common lines of research to achieve the revision of the present 8th edition with the objective to provide recommendations for changes to be introduce in the 9th edition. Special emphasis will be made in obtaining internal and external validation of new findings.

### 3.2. Primary Tumor

The prognostic relevance of tumor size has been thoroughly emphasized in the 7th and the 8th editions TNM. Every centimeter from 1 cm or less to 5 cm defines T categories with significantly different prognosis. Prognosis is significantly worse for tumors > 5 cm to 7 cm, and for those > 7 cm. This break down by tumor size has evolved over the years and is based on the increased quality of the registered data. If we had a larger amount of cases with detailed data, it is likely that even more categories could be established based on smaller fractions of size, but that would be impractical. Innovations may come from the analyses of specific T descriptors, like those defining T3. Special interest is focused on the invasion of the chest wall to find out if the involvement of its different layers—parietal pleura, bony structures, and soft tissue—has the same prognosis. No differences were found in the exploratory analyses of the 8th edition between parietal pleura invasion and deeper invasion of the chest wall, but the number of patients in each category was small-163 and 405, respectively-, and their 5-year survival rates were 56 and 52%, respectively, for completely resected N0M0 tumors [8]. However, others have found different results. Sakakura et al. compared the postoperative survival of 101 patients with T3 tumors with invasion of the parietal pleura alone with that of 102 patients with deeper involvement of the chest wall. Their 5-year survival rates were 50 and 36.7%, respectively (*p* = 0.028). Based on their results, they proposed to subdivide the 8th edition T3 category into T3a (invasion of parietal pleura, only) and T3b (invasion of deeper structures of the chest wall) [14]. So, it is clear that this topic deserves further research. In addition, the Lepidic and AIS Subcommittee and the Imaging Subcommittee are working in conjunction with the T Subcommittee to validate Tis(AIS), T1mi, and lepidic predominant adenocarcinomas, and to confirm that the invasive part of part-solid adenocarcinomas leads prognosis. Clearer instructions will be given to improve the quality of data collections, and there are plans to publish an atlas to facilitate the diagnosis and the classification of these lesions.

### 3.3. Nodal Spread

The N Subcommittee and the Lymph Node Chart Subcommittee will further explore the possibility to add some type of quantification to the present N descriptors, based exclusively on the anatomic location of the lymph nodes. This is no straight forward task because quantifying nodal disease is easy at pathologic staging if an adequate lymphadenectomy has been performed, but not at clinical staging as mentioned in point 2.5. The limits of some nodal stations will be clarified, because some are simply marked by a straight line on the lymph node map, but are curved in reality, like the upper margin of the left pulmonary artery, which is the lower limit of the subaortic nodal station. The lower border of the azygos vein separates the right inferior paratracheal and the right hilar nodes, and its variable position poses difficulties when classifying these nodes, because some nodes defined as hilar are in fact in the mediastinum. Plans to add more anatomic drawings, intraoperative pictures and even videos to the present map or to an improved version of it are being discussed. There also is a particular interest in exploring the prognostic relevance of pathologic nodal status after induction therapy (ypN) to elucidate, for example, whether pN0 or pN2 have the same prognosis as ypN0 or ypN2. This is relevant because those patients who had received induction therapy had been excluded from previous analyses of the N component.

### 3.4. Distant Metastases

Shortly after the publication of the IASLC SPFC proposals for the 8th edition regarding the M descriptors [18], an independent group from Portugal validated the three M1 subcategories, but also found that having one or two metastases in the same organ had the same prognosis. They also found that prognosis was significantly better in patients with multiple metastases in a single organ than in those with multiple metastases in several organs [39]. This means that the prognostic impact of the number and location of the metastases has to be further studied, as well as the volume of the metastatic lesions, which may be a limiting factor for some local therapies, such as radiotherapy, radiofrequency, microwave, or ultrasound ablation. The impact of driver mutations will have to be considered, too, as prognosis of stage IV has changed if metastatic tumors with targetable driver mutations are properly treated. Finally, it would be desirable that the new database used to inform the 9th edition TNM had the required detail to enable the validation of the definition of oligometastatic disease, defined by consensus in 2019 as the presence of a maximum of 5 lesions in three different organs, excluding lymph node, serosal, and bone marrow metastases [40].

### 3.5. Stage Grouping and Prognostic Groups

Tumors with different TNM classifications but similar prognosis have been traditionally grouped into stages by the UICC and the AJCC. The increasing granularity of the IASLC databases used to inform the 7th and the 8th editions TNM allowed the refinement of the stage grouping. In the 7th edition, there were more T categories based on tumor size, and some tumors changed stage, but there were no new stages compared with those of the 6th edition TNM [4]. For the 8th edition, more stages were needed. Stages IA1, IA2, and IA3 were necessary to accommodate the three new T1 categories, because, in the absence of nodal and metastatic spread, T1a, T1b, and T1c have statistically significant different prognosis. The new stage IIIC was created to include T3 and T4 tumors with N3 disease. Finally, stage IV was divided into IVA, to include M1a together with M1b, which have similar prognosis, and stage IVB to include M1c [6].

The above innovations meant a very small improvement in the *R*^2^, a statistical measure of the amount of prognosis explained by the stages. For the 7th edition, the *R*^2^ for clinical and pathologic stages was 67.5 and 45.7, respectively, while for the 8th edition, it was 68.3 and 46.9, respectively [6]. So, as an average, it could be said that stage grouping in lung cancer explains roughly 60% of prognosis. It is well known that prognosis does not depend on the anatomic extent of the tumor, only. There are other tumor-related factors, as well as patient-related and environmental-related factors that modify prognosis and that are not considered in the TNM classification. The challenge for the 9th edition TNM will be how to combine these factors to enhance the prognostic power of the TNM classification, so that a more personalized prognosis can be given to a particular patient.

The UICC and the AJCC hold different views on how this combination of factors is to be done. The UICC considers that staging, as a function of the T, the N, and the M components of the classification, should remain strictly anatomic, with no other parameters. However, the AJCC, in the 8th edition of its staging manual, introduced the concept of prognostic stage groups, in which anatomic and non-anatomic parameters are mixed. For example, TNM and age for differentiated thyroid carcinoma; TNM, prostate specific antigen and histologic grade for prostate cancer; or TNM, estrogen receptor expression, progesterone receptor expression, human epidermal growth factor receptor 2, and histological grade for breast cancer [41]. The UICC prefers the term prognostic groups when anatomic and non-anatomic prognostic factors are combined, reserving the term stage for the combination of tumors with similar prognosis classified according to their anatomic extent by the TNM classification [42,43]. In the 8th edition of the UICC manual, there are grids with different prognostic factors for most tumors. For lung cancer, there are specific prognostic factors for surgically resected and locally advanced or metastatic non-small cell lung cancer, and for small cell lung cancer. These are tumor-, patient-, and environment-related factors that are classified as essential, additional, and new and promising [43]. Although they do not change the TNM classification or the stage of the tumors, they are helpful in refining the prognosis for an individual patient and in making therapeutic decisions.

For the 9th edition, the items that are now being collected in the IASLC database include, as a novelty, biomarkers—genetic biomarkers, copy number alterations and protein alterations. With this tumor profile combined with the TNM classification and with relevant clinical and environmental factors, prognosis and therapy will be more individualized and improved over that provided exclusively by the anatomic extent of the tumor.

### 3.6. Essential TNM

The UICC with the International Agency for Research in Cancer and the National Cancer Institute developed a classification system called Essential TNM designed to determine tumor stage when complete information is not available. At the moment, there are essential TNMs for breast, cervix, colon, and prostate cancers [43]. The IASLC SPFC is now evaluating two possible schemas of Essential TNM for lung cancer that will be discussed along the year. The chosen one will be presented to the UICC and, if accepted, will appear in the 9th edition of the TNM classification for lung cancer.

The way to define the Essential TNM is to determine with the most specific available information whether there are distant metastasis or not. If there are, the tumor is classified as stage IV (distant disease) and there is no need to further investigate. If there are no metastases, then the involvement of the loco-regional lymph nodes is investigated and their extent (extensive or limited) assessed. If the nodes are involved, it is stage III (regional disease) and no further investigations are done. If they are not involved, then the size and the extent of the primary tumor is studied and classified as advanced or limited, stage II or I, respectively (localized disease). So, the Essential TNM is determined in reverse order compared with the standard TNM, in which the primary tumor followed by the loco-regional nodal involvement and the distant metastases are usually investigated [44]. The Essential TNM is meant, mainly, for underdeveloped and developing countries with scarce resources to facilitate the classification of malignant tumors with the use of as few diagnostic tests as possible.

## 4. Conclusions

The periodic revisions of the TNM classification for lung cancer have progressively improved our understanding of the prognostic impact of the anatomic tumor extent. The innovations introduced in 7th and the 8th editions were based on large international databases, and increased our understanding of the prognostic impact of tumor size, the amount of nodal disease, and the number and location of distant metastases.

For the forthcoming 9th edition of the classification, an important objective will be the enhancement of prognostication by combining the TNM classification with other tumor-, patient-, and environment-related prognostic factors in prognostic groups. To this end, the registration of biomarkers will be essential to characterize tumor profile and to refine prognosis and therapy in a more personalized way. International collaboration will be of paramount importance to build a large and detailed database to achieve the proposed objectives.

## Figures and Tables

**Table 1 cancers-13-01940-t001:** Timeline of the TNM classification for lung cancer.

Year	Event or Edition
1943–1952	Pierre Denoix presented the TNM classification
1960–1967	The UICC published TNM brochures
	The one for lung cancer in 1966
1968	1st edition UICC TNM
1975	2nd edition UICC TNM
1977	1st edition AJCC TNM
1978	3rd edition UICC TNM, revised in 1982
1983	2nd edition AJCC TNM
1987	4th edition UICC TNM
1988	3rd edition AJCC TNM
1992	4th edition AJCC TNM
1997	5th edition UICC and AJCC TNM
2002	6th edition UICC and AJCC TNM
2009	7th edition UICC and AJCC TNM
2016	8th edition UICC and AJCC TNM

TNM: tumor, node and metastasis classification; UICC: Union for International Cancer Control; AJCC: American Joint Committee on Cancer.

**Table 2 cancers-13-01940-t002:** Proposed nodal categories based on the number of involved nodal zones, 7th edition TNM.

Proposed Categories	Descriptor	Median Survival (Months)	5-Year Survival	Comparison	Hazard Ratio	*p* Value
N1a	Single N1 zone involved	52	48%			
N1b	Multiple N1 zones involved	31	35%	vs. N1a	1.32	<0.0090
N2a	Single N2 zone involved	35	34%	vs. N1b	1.04	0.7137
N2b	Multiple N2 zones involved	19	20%	vs. N2a	1.65	<0.0001

**Table 3 cancers-13-01940-t003:** Proposed nodal categories based on the number of involved nodal stations, 8th edition TNM. Survival is shown for the group of patients with complete resections, adjusted for histological type (adenocarcinoma vs others), sex, age > 60 years, and region of data origin.

Proposed Categories	Descriptor	Median Survival (Months)	5-Year Survival	Comparison	Hazard Ratio	*p* Value
N1a	Single N1 station involved	Not reached	59%			
N1b	Multiple N1 stations involved	60.9	50%	vs N1a	1.39	0.0005
N2a1	Single N2 station involved without N1	70.9	54%	vs N1b	0.89	0.2863
N2a2	Multiple N2 stations involved with N1	46.0	43%	vs N2a1	1.36	0.0007
N2b	Multiple N2 stations involved	40.0	39%	vs N2a2	1.26	0.0026

**Table 4 cancers-13-01940-t004:** Classification of lung cancers with multiple lesions.

Pattern of Disease	Recommended Classification
Multiple primary tumors	One TNM for each tumor
Separate tumor nodules	T3 if the nodules are in the same lobe of the primary tumor
	T4 if in another ipsilateral lobe
	M1a if in the contralateral lung
Multiple adenocarcinomas with ground glass/lepidic features	Highest T followed by the number of lesions or ‘m’ for ‘multiple’ in parenthesis, and one N and one M for all tumors: highest T(#/m)NM
Pneumonic type adenocarcinoma	If the tumor is measurable, it follows the general rules to assign a T category based on tumor size
	If the tumor is not measurable, T3 if it involves one lobe; T4 if it involves another ipsilateral lobe; and M1a if it involves both lungs

## Data Availability

No new data were created or analyzed in this study. Data sharing is not applicable to this article.

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
