# Peer review of "Future Perspectives on the TNM Staging for Lung Cancer"

_cancers, 2021, doi:10.3390/cancers13081940_

Round 1

Reviewer 1 Report

I thank the Editor for the opportunity of reviewing this paper

I commend  the authors for their valuable work 

I have only a minor comment regarding the following condition: tumor with adjacent lobe invasion.

This tumor is classified as T2  and the TNM classifications proposed in the years (including the last 8th edition by IALCS) do not differ the prognosis of tumor with adjacent lobe invasion in relation to fissure status (complete or incomplete fissure). 

Do the authors believe that a reclassification of this type of tumor is needing also in the light of fissure status ? The prognosis could be changed if the fissure are complete or incomplete. Thus, do the status of fissure guide the extend of resection of the adjacent affected lobe (lobar resection, segmentectomy, wedge resection ?)    

Author Response

Response to Reviewer 1 Comment

Point 1: I thank the Editor for the opportunity of reviewing this paper.

I commend the authors for their valuable work.

Response 1: Thank you very much for your kind comment.  

Point 2: I have only a minor comment regarding the following condition: tumor with adjacent lobe invasion.

This tumor is classified as T2 and the TNM classifications proposed in the years (including the last 8th edition by IALCS) do not differ the prognosis of tumor with adjacent lobe invasion in relation to fissure status (complete or incomplete fissure). 

Do the authors believe that a reclassification of this type of tumor is needed also in the light of fissure status? The prognosis could be changed if the fissures are complete or incomplete. Thus, do the status of fissure guide the extend of resection of the adjacent affected lobe (lobar resection, segmentectomy, wedge resection?) 

Response 2: Thank you very much for this valuable comment. Tumor invasion of the adjacent lobe poses two kinds of problems: one concerns the proper classification according to the TNM system; the other concerns the most appropriate resection of these tumors.

Regarding the classification of a lung cancer with invasion of the adjacent lobe, there is evidence for supporting the T2 category and the T3 category. Because the results are conflicting, General Rule number 4 of the TNM classification is applied: “If there is doubt concerning the correct T, N, or M category to which a particular case should be allotted, the lower (i.e., less advanced) category should be chosen. This will also be reflected in the stage”. (Brierley JD, Gospodarowicz MK, Wittekind C, eds; UICC TNM Classification of Malignant Tumours, 8th edition,Oxford, UK: Wiley Blackwell, 2017, page 4).

The status of the fissure has prognostic implications. If the fissure is complete and the tumor invades the adjacent lobe, prognosis is worse than in tumors that invade the adjacent lobe but the fissure is incomplete. The latter has been proposed for upstaging to T2b if tumors are 5cm or less in greatest dimension (Ohtaki Y et al. Eur J Cardiothorac Surg 2013; 43:302-309).

Regarding the proper type of resection of these tumors, several factors should be considered: the patient’s capacity to undergo a bilobectomy or a pneumonectomy, and the magnitude of the invasion of the adjacent lobe, which may be managed by a wedge, a segmentectomy or a lobectomy. Data on this topic are scarce, but a recent study suggests that and additional sublobar resection might be enough for tumor with adjacent lobe invasion if the fissure is incomplete (Andreetti C et al. Thoracic Cancers 2020; 11: 232-242).

All the above is very relevant, so three new paragraphs and more references have been added to section 2.2. Prognostic relevance of visceral pleura invasion of the revised text to explain all these issues. 

Reviewer 2 Report

Thank you for inviting me to review this paper. This article outlines the history and future perspectives of the revision of the TNM classification of lung cancer by the famous Dr Ramón Rami-Porta. I would like to congratulate Dr. Ramón Rami-Porta for his work on the TNM classification over the years. I have few comments to make on this article and consider it to be an acceptable report in its present form.

If I may make any comment, as described in this article, the categorization of the degree of pleural invasion (PL1/PL2) of the T2a tumors and its reflection on the T factor, as well as the subcategorization of PL3 lung cancers involving neighboring structures based on the depth of invasion (only pleura/other deeper invasion) may be one of the future considerations for the 9th edition of the TNM classification of lung cancer.

As the author described, prognostic difference between the T2a-PL1 and T2a-PL2 tumor groups may be significant, which indicates that the T2a-PL2 group can be categorized into a worse prognostic category (T2b). Alternatively, among PL3 tumors involving neighboring organs, subcategorization of whether only the pleura was infiltrated (T3a) or other deeper structures were invaded (T3b) could be a practical consideration [1].

[1] Sakakura N, Mizuno T, Kuroda H, Arimura T, Yatabe Y, Yoshimura K, Sakao Y. The eighth TNM classification system for lung cancer: A consideration based on the degree of pleural invasion and involved neighboring structures. Lung Cancer. 2018;118:134-138.

It would be appreciated if the above considerations and the report could be briefly mentioned in order to suggest one possibility of direction for future revision of the TNM classification, if possible.

This report, as well as the previous works of the author, will be of great interest and suggestiveness to many readers. It may be superfluous to add a comment to this excellent perspective. I hope my comments will be of some help.

Thank you again for the opportunity to review this excellent work.

Author Response

Response to Reviewer 2 Comment

Point 1: Thank you for inviting me to review this paper. This article outlines the history and future perspectives of the revision of the TNM classification of lung cancer by the famous Dr. Ramón Rami-Porta. I would like to congratulate Dr. Ramón Rami-Porta for his work on the TNM classification over the years. I have few comments to make on this article and consider it to be an acceptable report in its present form.

Response 1: Thank you very much. I highly appreciate your kind comment.

Point 2: If I may make any comment, as described in this article, the categorization of the degree of pleural invasion (PL1/PL2) of the T2a tumors and its reflection on the T factor, as well as the subcategorization of PL3 lung cancers involving neighboring structures based on the depth of invasion (only pleura/other deeper invasion) may be one of the future considerations for the 9th edition of the TNM classification of lung cancer.

As the author described, prognostic difference between the T2a-PL1 and T2a-PL2 tumor groups may be significant, which indicates that the T2a-PL2 group can be categorized into a worse prognostic category (T2b). Alternatively, among PL3 tumors involving neighboring organs, subcategorization of whether only the pleura was infiltrated (T3a) or other deeper structures were invaded (T3b) could be a practical consideration [1].

[1] Sakakura N, Mizuno T, Kuroda H, Arimura T, Yatabe Y, Yoshimura K, Sakao Y. The eighth TNM classification system for lung cancer: A consideration based on the degree of pleural invasion and involved neighboring structures. Lung Cancer. 2018;118:134-138.

It would be appreciated if the above considerations and the report could be briefly mentioned in order to suggest one possibility of direction for future revision of the TNM classification, if possible.

Response 2: There are two important issues in this comment: first, the different prognosis of PL1 and PL2; and second, the different prognosis of the depth of invasion of the chest wall.

The differences in PL1 and PL2 are addressed in section 2.2. Prognostic relevance of visceral pleura invasion of the revised text, in reply to the comment of Reviewer 1. In addition, a new paragraph has been added to this section to explain the prognostic importance of the status of the fissures, complete or incomplete.

The relevance of the depth of invasion of the chest wall structures was briefly covered in section 3.2 Primary tumor of the manuscript, indicating that, in the IASLC database, no significant differences were found in the 5-year survival of invasion of the parietal pleura alone and that of the invasion of deeper structures of the chest wall. However, it is true that other studies have found different results. Therefore, new text has been added to that paragraph and the reference kindly provided by Reviewer 2 has been duly cited.

Point 3: This report, as well as the previous works of the author, will be of great interest and suggestiveness to many readers. It may be superfluous to add a comment to this excellent perspective. I hope my comments will be of some help.

Thank you again for the opportunity to review this excellent work.

Response 3: I greatly appreciate your kind and encouraging words. Thank you very much.